

# Poly(ADP-ribosylation) is present in murine sciatic nerve fibers and is altered in a Charcot-Marie-Tooth-1E neurodegenerative model

Laura I. Lafon Hughes[1], Carlos J. Romeo Cardeillac[2], Karina B. Cal Castillo[2], Salomé C. Vilchez Larrea[3], José R. Sotelo Sosa[2], Gustavo A. Folle Ungo[1], Silvia H. Fernández Villamil[3,4] and Alejandra E. Kun González[2,5]

[1] Departamento de Genética, Instituto de Investigaciones Biológicas Clemente Estable (IIBCE), Montevideo, Uruguay
[2] Departamento de Proteínas y Acidos Nucleicos, Instituto de Investigaciones Biológicas Clemente Estable, Montevideo, Uruguay
[3] Instituto de Investigaciones en Ingeniería Genética y Biología Molecular "Dr. Héctor N. Torres, Consejo Nacional de Investigaciones Científicas y Técnicas, Buenos Aires, Argentina
[4] Departamento de Química Biológica, Facultad de Farmacia y Bioquímica, Universidad de Buenos Aires, Buenos Aires, Argentina
[5] Departamento de Biología Celular y Molecular, Sección Bioquímica, Facultad de Ciencias, Universidad de la República, Montevideo, Uruguay

Corresponding authors
Laura I. Lafon Hughes, lauralafon2010@gmail.com
Alejandra E. Kun González, akun@fcien.edu.uy

## ABSTRACT

**Background.** Poly-ADP-ribose (PAR) is a polymer synthesized by poly-ADP-ribose polymerases (PARPs) as a postranslational protein modification and catabolized mainly by poly-ADP-ribose glycohydrolase (PARG). In spite of the existence of cytoplasmic PARPs and PARG, research has been focused on nuclear PARPs and PAR, demonstrating roles in the maintenance of chromatin architecture and the participation in DNA damage responses and transcriptional regulation. We have recently detected non-nuclear PAR structurally and functionally associated to the E-cadherin rich *zonula adherens* and the actin cytoskeleton of VERO epithelial cells. Myelinating Schwann cells (SC) are stabilized by E-cadherin rich autotypic *adherens junctions (AJ)*. We wondered whether PAR would map to these regions. Besides, we have demonstrated an altered microfilament pattern in peripheral nerves of Trembler-J (Tr-J) model of CMT1-E. We hypothesized that cytoplasmic PAR would accompany such modified F-actin pattern.
**Methods.** Wild-type (WT) and Tr-J mice sciatic nerves cryosections were subjected to immunohistofluorescence with anti-PAR antibodies (including antibody validation), F-actin detection with a phalloidin probe and DAPI/DNA counterstaining. Confocal image stacks were subjected to a colocalization highlighter and to semi-quantitative image analysis.
**Results.** We have shown for the first time the presence of PAR in sciatic nerves. Cytoplasmic PAR colocalized with F-actin at non-compact myelin regions in WT nerves. Moreover, in Tr-J, cytoplasmic PAR was augmented in close correlation with actin. In addition, nuclear PAR was detected in WT SC and was moderately increased in Tr-J SC.
**Discussion.** The presence of PAR associated to non-compact myelin regions (which constitute E-cadherin rich autotypic *AJ*/actin anchorage regions) and the co-alterations experienced by PAR and the actin cytoskeleton in epithelium and nerves, suggest

that PAR may be a constitutive component of *AJ*/actin anchorage regions. Is PAR stabilizing the *AJ*-actin complexes? This question has strong implications in structural cell biology and cell signaling networks. Moreover, if PAR played a stabilizing role, such stabilization could participate in the physiological control of axonal branching. PARP and PAR alterations exist in several neurodegenerative pathologies including Alzheimer's, Parkinson's and Hungtington's diseases. Conversely, PARP inhibition decreases PAR and promotes neurite outgrowth in cortical neurons *in vitro*. Coherently, the PARP inhibitor XAV939 improves myelination *in vitro*, *ex vivo* and *in vivo*. Until now such results have been interpreted in terms of nuclear PARP activity. Our results indicate for the first time the presence of PARylation in peripheral nerve fibers, in a healthy environment. Besides, we have evidenced a PARylation increase in Tr-J, suggesting that the involvement of cytoplasmic PARPs and PARylation in normal and neurodegenerative conditions should be re-evaluated.

## INTRODUCTION

*Adherens junctions* are protein complexes localized at intercellular junctions, characterized by the existence of a link to the actin cytoskeleton at their cytoplasmic face. The central molecules in these junctions are transmembrane proteins called cadherins. While extracellularly bound to a neighbor identical molecule, cadherins are intracellularly attached to several proteins that allow the anchorage of the actin microfilaments. In polarized epithelial cells, *adherens junctions* are grouped. They describe a band across the lateral cell faces which encircles the cell, called the *zonula adherens* or the adhesion belt, usually more basal than tight junctions (*Alberts et al., 2002*; *Meng & Takeichi, 2009*). Epithelial *adherens junctions* functions are not only structural but also regulatory, participating in cell signaling networks. For example, some *adherens junctions* proteins such as β-catenin are called NACos (Nuclear and Adherent junction Complex components). NACos can either localize at the *adherens junctions* or translocate to the nucleus where they act as transcription factors, allowing the coordination of the loss of cellular adhesion with cell rounding and mitosis entrance (*Pérez-Moreno, Jamora & Fuchs, 2003*; *Cerejido, Contreras & Shoshani, 2004*).

Clustered *adherens junctions* can also be found in other polarized cells. For example in the vertebrate peripheral nervous system, each axon is surrounded by Schwann cells and an extracellular matrix. These components altogether constitute the nervous fiber, characterized by longitudinal as well as radial polarity. Myelinating Schwann cells wrap around the axons of motor and sensory neurons to form the myelin sheath. E-cadherin autotypic junctions contribute to the overall stability of the Schwann cell, being located in all the autotypic junctions regions, namely the outer and inner mesaxons, the outer and inner loops, the paranodal regions and the Schmidt-Lanterman incisures (*Fannon et al., 1995*; *Poliak et al., 2002*).

Interestingly, Trembler-J (Tr-J) mice harbour a punctual missense mutation in peripheral myelin protein 22 (pmp-22) gene, constituting a model of hypomyelinating human peripheral neuropathy Charcot-Marie-Tooth-1E (CMT1-E, *Li et al., 2013*; formerly classified as CMT1-A; *Valentijn et al., 1992*). Tr-J mice display an increased number of Schwann cells (*Robertson et al., 1997*) as well as structural alterations in Schmidt Lanterman incisures and paranodes of their sciatic nerve fibers involving changes in autotypic *adherens junctions* components such as E-cadherin (*Devaux & Scherer, 2005*). Moreover, actin microfilaments abundance and distribution in sciatic nerve fibers are distorted (*Kun et al., 2012a*).

Poly-ADP-ribose (PAR) is a linear or branched polymer of up to 400 ADP-ribose units. It is synthesized by poly-ADP-ribose polymerases (PARPs) from $NAD^+$ as a postranslational protein modification and catabolized mainly by the endo- or exo-glicosidic action of poly-ADP-ribose glycohydrolase (PARG) (*Virag & Szabó, 2002*). PAR can in turn interact non-covalently with different protein domains including PBZ (PAR binding zinc finger motif), WWE (with three conserved residues -tryptophans and glutamate-) and Macrodomain (*Leung, 2014*).

Excessive nuclear PARP and PAR occur in brains in the context of neurodegeneration for example in Alzheimer's, Parkinson's and Hungtington's diseases (*Love, Barber & Wilcock, 1999*; *Martire, Mosca & d'Erme, 2015*; *Vis et al., 2005*; *Cardinale et al., 2015*). Besides, PAR quantity and/or PARP expression are altered in several other pathologies (*Strosznajder, Jesko & Strosznajder, 2000*; *Virag & Szabó, 2002*; *Strosznajder, Jesko & Zambrzycka, 2005*; *Masutani, Nakagama & Sugimura, 2005*; *Lafon-Hughes et al., 2008*; *Cerboni et al., 2010*; *Strosznajder et al., 2012*; *Liu et al., 2014*). Nevertheless, poly-ADP-ribosylation (PARylation) biology is still poorly understood.

Human PARP family has 17 members, 4 of which have PARylating activity. They are PARP-1, PARP-2, tankyrase-1 (TNKS-1) and tankyrase-2 (TNKS-2; *Vyas et al., 2005*). In spite of the fact that PARP-1 is the only member that is localized exclusively in the nucleus and PARG nuclear and cytoplasmic isoforms have been described, most PARylation studies are focalized in the cell nucleus. Nuclear PARPs regulate chromatin structure and somehow participate in nuclear networks regulating DNA replication, gene expression, DNA damage recognition and repair or telomere maintenance (*Virag & Szabó, 2002*). More recently, cytoplasmic roles of PARylation are being envisaged (*Lehtio, Chi & Krauss, 2013*; *Vyas et al., 2013*). For example, in epithelial cells TNKS-1 maps not only to the nucleus but also to endoplasmic reticulum, Golgi apparatus, secretion vesicles, lisosomes or epithelial lateral membrane. Moreover, TNKS-1 is recruited from the cytoplasm to the epithelial lateral membrane upon formation of E-cadherin-based *adherens junctions* (*Yeh et al., 2006*). *Adherens junctions* proteins vinculin and catenin have been recovered as PARylated proteins in immunoprecipitation experiments (*Gagné et al., 2008*; *Gagné et al., 2012*). Furthermore, VERO cells (green monkey renal epithelial cells) harbor a PAR belt associated to the epithelial adhesion belt which is synthesized during the cell–cell adhesion process. If actin polymerization is inhibited, the PAR belt is disaggregated. Conversely, if PAR belt synthesis is inhibited by the TNKS inhibitor XAV939, the actin cytoskeleton, cell shape

and cell adhesion are altered, indicating that the PAR belt is structurally and functionally linked (directly or indirectly) to the actin cytoskeleton (*Lafon-Hughes et al., 2014*).

With the aim of contributing to understanding the biology of PARylation in the peripheral nervous system, we tested the following three hypothesis. First, as PAR exists associated to epithelial *adherens junctions* in VERO cells, it might be also found in peripheral nervous system Schwann cell non-compact myelin which is rich in autotypic *adherens junctions*. Second, as PAR is associated to the actin cytoskeleton in VERO cells and F-actin is highly increased in Tr-J sciatic nerve fibers, PAR might be more abundant in Tr-J than WT nerves. Third, as nuclear PAR increases have been described in central neuropathology, nuclear Schwann cell-PAR could be increased in neurodegenerative Tr-J peripheral nerve fibers.

In the present work, we have used immunohistofluorescence (IHF) and confocal microscopy to evidence for the first time the presence of PAR in Schmidt Lanterman incisures and paranode regions in WT sciatic nerves. The non compact myelin regions where PAR and actin colocalized mimicked E-cadherin well-described distribution. Besides, we demonstrated through filamentous actin (F-actin) and PAR signals quantification on cytoplasmic axonal and Schwann cell domains that Tr-J mice sciatic nerves have increased PAR.

## MATERIALS AND METHODS

### Mice sciatic nerves

Institutional and national guidelines for the care and use of laboratory animals were followed. All animal procedures were performed following the recommendations of the Committee of Ethics in the Use of Animals Comité de Etica en el Uso de Animales (CEUA)-IIBCE, approved experimental protocols 011/11/2014 and 002/05/2016. Male 70 to 90 days old (P70-P90) wild-type (WT) and heterozygous mice carrying a mutation in *pmp-22* (Tr-J) from Jackson Laboratory (strain B6.D2-Pmp22 Tr-J/J) were killed by cervical dislocation. Sciatic nerve dissection was promptly carried out in less than one minute and followed by fixation through immersion in cold freshly prepared 3% w/v paraformaldehyde (PFA) in PHEM buffer (60 mM Pipes, 25 mM Hepes, 10 mM EGTA, 2 mM magnesium chloride, pH 7.2-7.6) for 1 h. This procedure is known to grant a faster fixation than systemic descendent perfusion in the singular case of mice sciatic nerves (*Kun et al., 2012a*; *Kun et al., 2012b*). Then, nerves were cryoprotected in sucrose/PHEM at 4 °C (increasing concentrations along 24 h: 5% to 30% w/v) (*Kun et al., 2012a*; *Kun et al., 2012b*). Next, tissue infiltration was done through progressive substitution (25%, 50%, 100%) of sucrose/PHEM by Jung Tissue Freezing Medium (Leica 0201 08926). Cryosections (10 μm) were cut using a Cryostat (SLEE) and adhered to slides precoated with chromic gelatin. Optimum results were obtained when overnight cryoprotection followed by infiltration, embedding, freezing and cutting were done within 30 h. Cryosections were stored at −20 °C and immunostaining was started as soon as possible, in the following 24 h.

### Immunostaining with BD anti-PAR antibody

Before immunostaining, sciatic nerve sections were air-dried for 10 min at room temperature (RT) and incubation chambers were delimited on slides with nail polish.

Sections were washed in filtered PHEM (fPHEM), postfixed in 2% w/v PFA/fPHEM for 15 min, washed in fPHEM, and permeabilized in 0.1% v/v Triton-X100/fPHEM for 30 min. Free aldehydes were blocked with 1% w/v sodium borohydride (NaBH4, Fluka 71320) in fPHEM (10 min). Then, tissue sections were washed in PHEM and immersed for 30 min in blocking buffer [100 mM L-Lysine (SIGMA L5501), 0.1% w/v bovine seroalbumin (BSA, SIGMA A-2153) and 5% v/v goat serum in PHEM]. Sciatic nerve cryosections were incubated with 1:200 Becton Dickinson rabbit anti-PAR (BD 551813) for 2 h at 37 °C diluted in incubation buffer [200 mM L-lysine and 0.1% w/v BSA in PHEM]. After washing in fPHEM, sections were incubated for 1 h at RT in blocking buffer with goat anti-rabbit secondary antibodies conjugated to Alexa 488 (Invitrogen A11034) and an Alexa Fluor 546-phalloidin probe (Molecular Probes TM A22283) to evidence filamentous actin (F-actin). Finally, after nuclear counterstaining with DAPI (1.5 µg/mL/fPHEM; Molecular Probes TM D21490), slides were rinsed in fPHEM, mounted in Vectashield (Vector 94010) or Prolong Gold (Molecular Probes P36930) and sealed with nail polish.

Controls without primary antibody were always run in parallel to check the specificity of the detected signals. Moreover, to assure that specifically in sciatic nerves the antigen detected was PAR, two complementary approaches were undertaken.

## BD anti-PAR antibody validation in sciatic nerve. I: alternative anti-PAR antibody

ENZO BML-SA216 anti-PAR antibody was labelled using a kit (CFTM 488A) following the manufacturer's instructions. Then, direct IHF was performed.

## BD anti-PAR antibody validation in sciatic nerve. II: PAR digestion on fixed tissue sections

Sciatic nerve sections were subjected to digestion with recombinant human PARG (SIGMA SRP8023 lot A00634/A, containing 2 µg of PARG in 10 µL buffer). PARG effect was studied following ENZO recombinant PARG protocols (ALX-202-045-UC01), in a reaction buffer essentially described by *Ménard & Poirier (1987)*, used by *Thomassin et al. (1990)*, *Thomassin et al. (1992)* and *Brochu, Shah & Poirier (1994)* [50 mM potassium phosphate buffer pH 7.5, 50 mM KCl, 10 mM β-mercaptoethanol, 10% v/v glycerol, 1 mM DTT and 0.1% v/v Triton-X100]. Fixed sciatic nerve slices were rinsed in PARG reaction buffer and then incubated at RT with or without 50 nM PARG in 100 µL reaction buffer for 24 h. Then, indirect IHF with BD anti-PAR antibody was performed following the above described protocol.

## Confocal microscopy and image analysis

Image stacks were collected with an OLYMPUS FV300 or a Leica TCS SP5 II confocal microscope using 40X dry objectives (only overviews), a Plan Apo 60X/1.42 NA, a Plan Apo 63X/1.4 NA or a Plan Apo 100X/1.4 NA oil immersion objectives, with or without digital zoom. To assure signal specificity, original images were taken in the same conditions as reference images of cells not labeled with primary antibodies, at the same confocal session.

Nerve cutting ends were skipped (either excluding them before cryoprotection or avoiding the tips throughout image collection and quantification).

All images were processed and analyzed using Image J free software. Then data were exported to Microsoft Excel (Office 2016 Home and Student Microsoft 79G-04351).

Olympus Fluoview images were opened using UCSD/Fluoview control plugin. LEICA (.lif) files were extracted and saved as .tiff. The images used for illustrations were processed adjusting brightness/contrast in parallel in WT, TrJ and without primary antibody images to avoid artifacts, and finally smoothed (ratio1 Gaussian blur).

16 animals were sacrificed to study sciatic nerve PARylation levels and distribution (14 of them in 7 paired experiments including one WT and one Tr-J sibling). We evaluated qualitatively 7 experiments and quantitatively 3 experiments. Figshare links to see original microscopy stacks, processed figures, ROIset examples, raw data, normalized data, summarized data and statistics can be found in Table S1.

"Blind" quantification was impossible, given the differences in nuclei number, actin and PAR distribution and fluorescence levels. These characteristics immediately revealed the genotype of the sample being quantified.

Relative quantitation of F-actin and PAR was done on crude images using data from 3 independent experiments with paired animals (WT and Tr-J brothers). Images were taken under identical conditions for WT and Tr-J in each experiment, taking as a reference the control without primary antibody.

To evaluate cytoplasmic relative contents, F-actin (red) and PAR (green) signal strength was measured on at least 350 DAPI-negative regions of interest (ROIs) marked through the fiber diameter of 12 stacks per experimental condition. The length of the ROI was normalized and divided into 10 equal parts. In each experiment, the average relative intensity of actin and PAR signals along the fiber diameter was calculated. Besides, assuming that the intervals from 0 to 30% and from 70 to 100% correspond to Schwann cell (SC) regions whereas the range of 30 to 70% corresponds to axon regions (*Kun et al., 2012a*), it was possible to calculate the average relative intensity of F-actin and PAR in the axon. Then the average relative intensities of PAR and actin in the axon were assigned the value 1 and used to normalize all the measurements. Finally, normalized data from the three experiments were pooled. Mean intensities in arbitrary units (relative to axon WT), standard deviations and standard errors were calculated for F-actin and PAR signals in WT and Tr-J axons and SC. Statistical significance was tested using two-tailed Student's $t$-test, with $p < 0.001$.

The gross anatomical differences between the WT and TrJ nerves could have skewed the signal quantification. In order to exclude such a putative artifact, we present data (originally collected with a different purpose in the context of Karina Cal's Master Thesis, *2017*) showing that an unrelated signal, quantified following the same methodology (on ROIs through fiber diameters) did not increase. This signal corresponds to the cytoplasmic fraction of the octamer binding transcription factor-6 (OCT6), which is a nucleocytoplasmic shuttling protein according to *Baranek, Sock & Wegner (2005)*.

To assess relative nuclear PAR contents, the ImageJ plugin "Intensity_Ratio_Nuclei_Cytoplasm" was used. This allowed an automatic recognition of

nuclear borders and areas. To avoid biases due to differences in nuclear number or area, we measured the average nuclear PAR signal intensity. This was done in at least 8 stacks per condition from 3 independent experiments. Again, data were normalized according to mean nuclear PAR in each experiment and then were pooled. Mean intensities in arbitrary units (relative to nuclear WT PAR), standard deviations and standard errors were calculated. Statistical significance was tested using two-tailed Student's $t$-test, with $p < 0.001$.

The "Colocalization Analysis/Colocalization Highlighter" ImageJ plugin (P. Bourdoncle, Institute Jacques Monod, Service Imagerie, Paris, France) was used to highlight the cell regions where F-actin and PAR colocalize. The user arbitrarily establishes a minimum threshold for each channel. In this case, the threshold was experiment-dependent but WT, Tr-J and without primary antibody images were processed in the same way. The program highlights (in yellow/orange) the pixels where the intensity is above the threshold for both channels. Nothing was highlighted in the absence of PAR primary antibody (black image); this was just an image processing control.

## RESULTS

### WT mice sciatic nerves contained epitopes recognized by BD anti-PAR antibodies, particularly evident in Schmidt Lanterman incisures, paranodes and outer loop regions

Nerve cryosections were subjected to indirect IHF with rabbit anti-PAR antibody as well as F-actin detection with phalloidin-543 probe and nuclear counterstaining with DAPI.

Throughout the work, figures illustrate fluorochrome signals corresponding to PAR (detected with BD anti-PAR antibody) in green, F-actin in red and DAPI in blue. Figures 1A and 1B show single confocal slices where PAR signals (absent in the control without primary antibody) can be seen.

For further analysis, colocalization was defined as the spatial overlap (in a single pixel) of two signals (red and green, correspondent to F-actin and PAR channels), being each of the signals above a certain intensity threshold (fixed arbitrarily in an experiment-specific way).

When pixels where F-actin and PAR signals colocalized were highlighted in yellowish/orange color, appealing images were obtained. The F-actin-PAR colocalization regions (overviewed in Fig. 1C) drew the known non-compact myelin regions in sciatic nerves where E-cadherin has been localized, namely Schmidt Lanterman incisures (Figs. 1D–1I), outer loops and paranodes (Figs. 1K–1P). The observed distribution of F actin-PAR is in close agreement to the E-cadherin distribution reported by Fannon and collaborators (see Figure 5 of Fannon et al., 1995).

### F-actin increase in Tr-J mice sciatic nerves was paralleled by BD anti-PAR signal rise

Once PAR signal was detected in WT sciatic nerves, parallel experiments were carried in WT and Tr-J siblings. As can be seen in Figs. 2A–2F, alterations in the distribution of

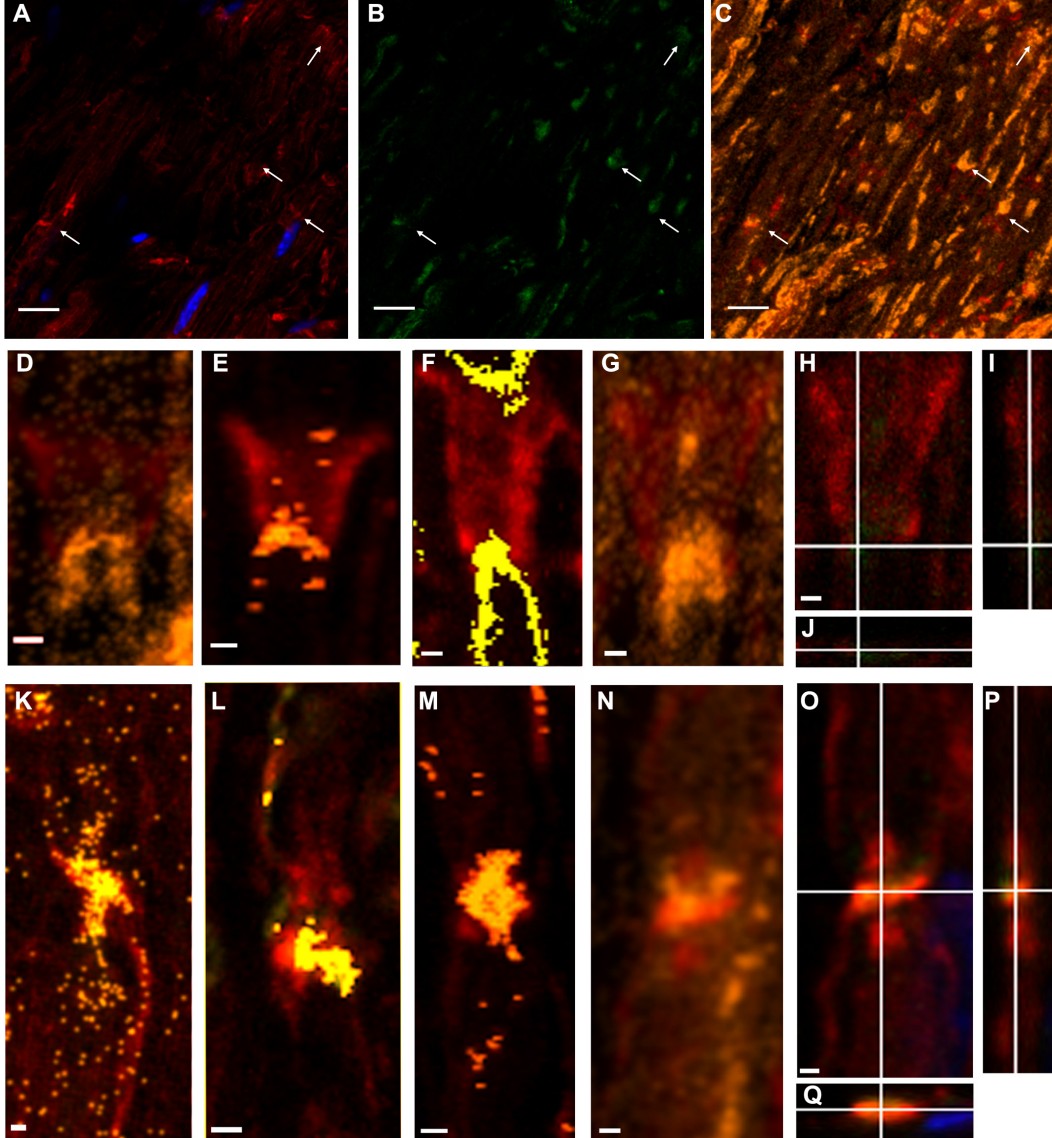

**Figure 1** **Poly-ADP-ribose (PAR) was present in WT sciatic nerves, particularly in non-compact myelin regions.** A conserved color code has been used in all the Figures. *Green:* PAR, *red:* F-actin; *blue:* DAPI; yellow/orange: F-actin-PAR colocalization highlight. All the images were obtained using BD anti-PAR antibody. (A–C): 100x sciatic nerve overview. (A, B) 100x single confocal slices; (C) correspondent F-actin-PAR colocalization highlight 3D projection. *Bar:* 10 μm. The arrows point to colocalization regions outlined in a single-plane which are better interpreted in the context of the 3D projection. Channel intensities could be enhanced to facilitate eye detection, but then the photographs would not be comparable to those in Fig. 3. (D–G) 3-D projections of Schmidt-Lanterman incisures extracted from (C) (D, G) and from other stacks. (H–J) XY, XZ and YZ cuts of the Schmidt-Lanterman incisures observed in (G). (K–N) 3-D projections of paranode regions extracted from (C) (N) and from other stacks. (O–Q) XY, XZ and YZ cuts of the paranodes observed in (N). *Bar:* 1 μm. This result was observed in 7 independent experiments.

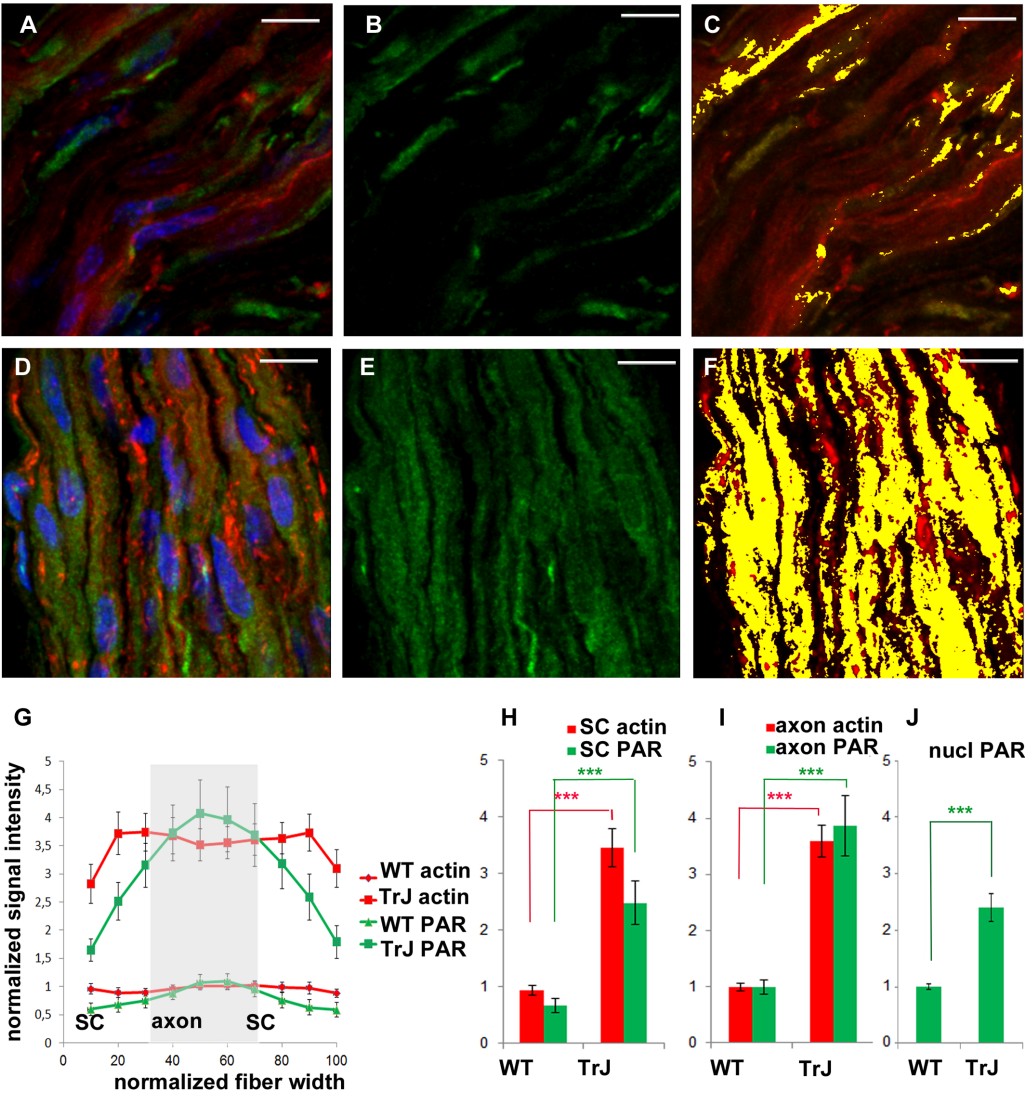

**Figure 2   F-actin increase inTr-J mice sciatic nerves was paralleled by PAR increase.** The color code
is maintained. *Green:* PAR, *red:* F-actin, *blue:* DAPI; *yellow*: colocalization highlighter mask. (A–C): WT
nerve; (D–F): Tr-J nerve. 3-D reconstructions from 100x confocal stacks. *Bar.* 15 µm. (G–I): relative
quantitation of cytoplasmic F-actin and PAR from 3 independent experiments with paired animals (WT
and Tr-J siblings). All PAR was detected with BD anti-PAR antibody. Confocal microscopy images were
taken under identical conditions for WT and Tr-J in each experiment, using as a reference the control
without primary antibody. F-actin (*red*) and PAR (*green*) signal strength were measured along the fiber
diameter in DAPI-negative ROIs. The length of the ROI was normalized and divided into 10 equal parts.
It was assumed that the intervals from 0 to 30% and from 70 to 100% correspond to Schwann cells (SC)
whereas the 30 to 70% range corresponds to axons. (G) Relative intensity of F-actin and PAR signals
along the normalized fiber diameter. Data were averaged by confocal stack ($n = 12$ WT and $n = 12$ Tr-J
stacks from three independent experiments). Then they were normalized by mean WT axon F-actin and
mean WT axon PAR of the corresponding experiment and expressed as stack mean ± s.e.m. (H): Relative
intensity of F-actin and PAR in the axons of WT (13,132 measurements) and Tr-J (9,409 measurements)
mice; $n = 12$ stacks. Mean ± s.e.m. (I) Counterpart of (H) in Schwann cells (WT: 8,483 measurements
and TrJ: 5983 measurements; $n = 12$ stacks). (J) Relative intensity of nuclear PAR in WT and Tr-J slices
($n = 55$ and 50 slices from 10 and 11 stacks respectively, from 3 independent experiments). Mean ± s.e.m.
(G–J) data were normalized by WT axonal actin and WT axonal PAR (G–I) or WT nuclear PAR intensity
(J). (***) $P < 0.001$ (Student's $t$ test). PAR increase was qualitatively observed in a total of 7 experiments.

PAR in Tr-J sciatic nerves were observed. A careful quantification of F-actin and PAR signals along the fiber diameters in DAPI-negative (cytoplasmic) fiber regions was done (Fig. 2G). Subsequently, a previously established structural criteria (*Kun et al., 2012a*), stating that roughly the inner 40% of the fiber diameter corresponds to the axon whereas the remnant outer region corresponds to the Schwann cell, was adopted. This assumption allowed to affirm that both in Schwann cells (Fig. 2H) and in axonal domains (Fig. 2I), PARylation increased coarsely three to four times, like F-actin, in Tr-J compared to WT nerves. Interestingly, an unrelated signal, corresponding to cytoplasmic OCT6, quantified following the same methodology (on ROIs through fiber diameters) did not increase (see Fig. S1), demonstrating that the observed elevation was not a quantification artifact.

Figures 3A–3C shows a Tr-J sciatic nerve from the same experiment as the WT nerve in Figs. 1A–1C. Besides, image processing was exactly the same; therefore, it is comparison-prone. Figures 3D–3I, comparable to Figs. 1N–1P, shows that the axon is the main region where F-actin and PAR colocalize in Tr-J sciatic nerves. An analogous phenomenon is observed in Figs. 3J–3O, depicting a Tr-J Schmidt Lanterman incisure comparable to WT Schmidt Lanterman incisures in Fig. 1D and 1G-I. In conclusion, PAR distribution in Tr-J sciatic nerve was altered even in still identifiable non-compact myelin regions.

Although this work was focused on cytoplasmic PARylation, nuclear PARylation was also detected. The average PAR signal in the nuclear area in Tr-J sciatic nerves was increased 2.5 times in relation to WT average nuclear signal.

### ENZO anti-PAR antibody but not Tulip clone H10 anti-PAR antibody reproduced the signals obtained with BD anti-PAR antibody in WT and Tr-J sciatic nerves

ENZO mouse monoclonal antibody, raised against short to mid-PAR chains (2 to 50 units) showed a clean signal after direct IHF (see Fig. S2). In fact, ENZO anti-PAR depicted clearly the paranodes, Schmidt Lanterman incisures and outer loops (Figs. SA, Figs. S2B) in WT sciatic nerves. An overview shows again that Tr-J sciatic nerves, which harbor extra nuclei (Fig. S2C vs Fig. S2F) as well as F-actin increase (D vs G), do also depict PAR increase (E vs H). The same can be perceived comparing I (WT) and J (Tr-J) merged channels images. Thus, ENZO anti-PAR antibody qualitatively conduces to the same conclusions as BD anti-PAR antibody (and this has been confirmed even quantitatively in nuclear WT vs Tr-J signals; see the Figshare link on Table S1).

In contrast, Tulip H10-clone antibody, known to recognize preferentially long branched PAR chains above 20 residues (*Kawamitsu et al., 1984*), showed no signal at all although it was assayed both in WT and Tr-J sciatic nerves (data not shown).

### BD anti-PAR signal was diminished after fixed sciatic nerves incubation in PARG

PARG effect was studied in a PARG reaction buffer (PARG-buff) essentially described by *Ménard & Poirier (1987)* which was later modified by *Thomassin et al. (1990)*, *Brochu, Shah & Poirier (1994)* and has given rise to ENZO recipe. PARG-buff was the control against which the digestion was evaluated.

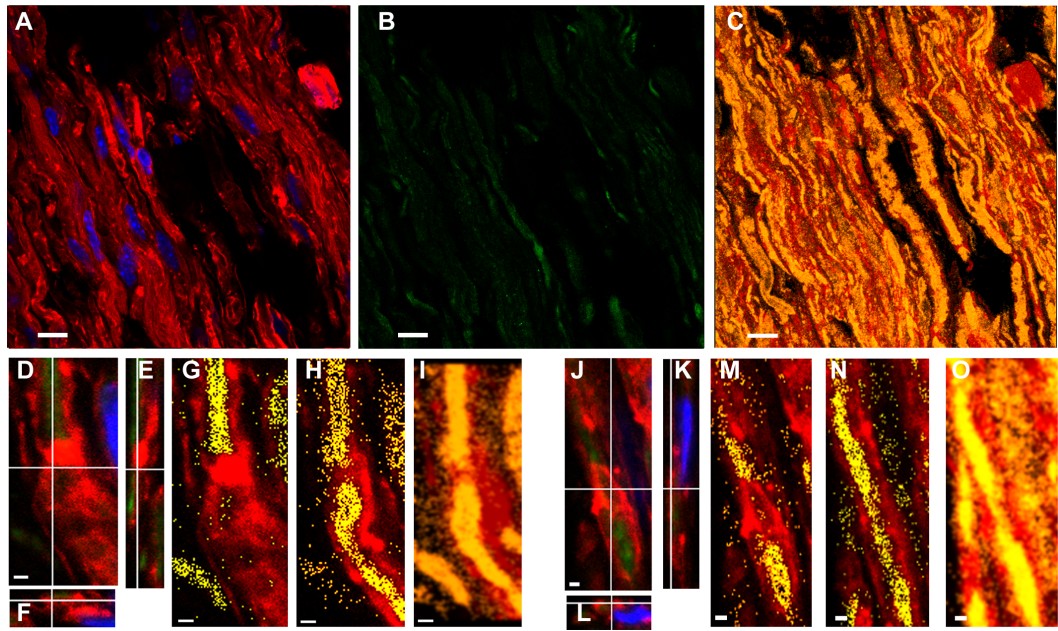

**Figure 3** **Even in still identifiable non-compact myelin regions of Tr-J sciatic nerves, the distribution of PAR was altered.** The color code is maintained. *Green:* PAR, *red:* F-actin, *blue:* DAPI; *yellow:* colocalization highlighter mask. All the images were obtained using BD anti-PAR antibody. (A–C) 100x Tr-J sciatic nerve overview [comparable to WT sciatic nerve from Figs. 1A–1C]. (A, B) 100x single confocal slices; (C) corresponding F-actin-PAR colocalization highlight 3D projection. *Bar:* 10 μm. (D–I) and (J–O) representative paranode-like and Schmidt-Lanterman-like structures extracted from (C). (D–F) XY, XZ and YZ cuts of a paranode. (G, H) single slices at different z-positions showing F- actin-PAR colocalization. (I) 3-D reconstruction of the paranode region. (J–L) XY, XZ and YZ cuts of a Schmidt-Lanterman. (M, N) single slices at different z-positions showing F-actin-PAR colocalization. (O) 3-D reconstruction of the paranode region. *Bar:* 1 μm. Altered PAR distribution was qualitatively observed in a total of 7 experiments.

To estimate an adequate PARG concentration, it was taken into account that a commercial PARG kit (Trevigen 4682-096-K) uses 5 ng of PARG in 100 μL to degrade PAR associated to 20 μg of protein extract in 30 min at RT. Besides, 6.75 ng of recombinant bovine PARG is enough to carry out a biochemical PARG degradation in an assay tube (see Fig. 6 from *Meyer et al., 2007*). Tissue digestion requires higher enzymatic concentrations, sometimes orders of magnitude higher, than biochemical "tube-reactions". For example, an RNAse concentration of 20 μg/mL is used in DNA extraction protocols (*Sambrook & Russell, 2001*) while a 500 times higher concentration (10 mg/mL) has been used to digest tissue RNA *in situ* (*Sotelo et al., 2013*). Nevertheless, as the amount of available enzyme was limited, escalating was not an option. Thus, 50 nM PARG in 100 μL PARG-buff was finally used in a 24 h digestion protocol. PARG effect on sciatic nerve fixed tissue sections was not homogeneous. We could observe tissue regions where nothing happened coexisting with huge or intermediate PARG effects. Huge effects means that PAR signal reached 10% in digested tissue relative to non-digested tissue (see Fig. S2K vs 1L). A clear difference could also be seen comparing non-digested with digested tissue in Fig. S2K vs L, N vs P or R vs T. Intermediate effects means partial digestion, like the PAR "cloud" observed in Fig. S2M. The reached PAR digestion in our borderline experimental conditions (PARG

concentration in the same order as in tube biochemical reactions, small digestion volume and uneven tissue surface) was a convincing result.

Interestingly, the F-actin levels seemed to diminish together with PAR levels at the digested regions (Figs. S2N–S2U).

To sum up, the evidence proves the PAR identity of IHF PAR signals, validating the biological findings, namely: (1) PAR colocalizes with with F-actin at non-compact myelin regions; (2) in a Charcot-Marie-Tooth demyelinating disease model in which sciatic nerves present an excess of F-actin, accompanied by E-cadherin delocalization (*Kun et al., 2012a*; *Devaux & Scherer, 2005*), detected PAR was excessive and was delocalized too; (3) nuclear PAR was detectable in WT and moderately increased in Tr-J sciatic nerve Schwann cells.

## DISCUSSION

This study represents the first report of PAR presence and distribution in the peripheral nerve system. We demonstrated that WT mice sciatic nerves contained nuclear PAR as well as cytoplasmic PAR. The latter colocalized with F-actin at E-cadherin rich non-compact myelin regions including Schmidt Lanterman incisures and paranodes.

Like in VERO epithelial belt, PAR was detected with anti-PAR antibodies that are presumed to recognize short to middle chain length polymer (BD and ENZO anti-PAR antibodies). The finding of PAR associated to E-cadherin rich/actin anchorage regions in two different biological systems (a monkey epithelial cell line and mice nerves) suggests that PARylation may be involved in *adherens junction* biology. These findings have deep implications in terms of junctional structures and cell signaling.

It has been proposed that inhibition of axonal regeneration by myelin after injury is an aberrant effect of an otherwise physiological inhibitory function (*Shen et al., 1998*). Axon branching inhibition is essential for the structural economy of the peripheral nervous system, especially for long axons. One key mediator of these functions would be myelin-associated glycoprotein (MAG; *Mukhopadhyay et al., 1994*; *Schafer et al., 1996*; *Shen et al., 1998*). MAG is mainly found at the adaxonal side of the Schwann cell membrane and non-compact myelin regions rich in autotypic *adherens junctions* including Schmidt Lanterman inscisures and paranodes (*Martini & Schachner, 1988*; *Ghabriel & Allt, 1980*; *Erb et al., 2006*).

In WT mice we observed PAR at non-compact myelin regions. Besides, PAR has been involved in axonal regeneration inhibition (*Brochier et al., 2015*). Therefore, an hypothesis can be raised regarding the putative physiological role of the observed PAR in preventing axonal branching.

Through image quantification of immunohistofluorescent images, we have also shown that Tr-J sciatic nerve Schwann cells harbored excessive nuclear PAR. Interestingly, excessive nuclear PARP and PAR had already been reported in brains in the context of neurodegeneration, for example, in Alzheimer's, Parkinson's and Hungtington's diseases (*Love, Barber & Wilcock, 1999*; *Martire, Mosca & d'Erme, 2015*; *Vis et al., 2005*; *Cardinale et al., 2015*). In fact, PARP-1-/- mice confirm the protective role of PARP-1 deficit towards injury induced by Aβ injections (that mimics Alzheimers disease), MPTP or 6-OHDA (used

to induce Parkinson-like symptoms). Thus, the current prevailing interpretations are that different injuries induce PARP-1 overactivation leading to cell death by energy shortage due to $NAD^+$ overconsumption and/or that DNA repair alterations (which may involve PARP-1 signaling) represent a common denominator in neurodegeneration (*Martire, Mosca & d'Erme, 2015*; *Ross & Truant, 2016*). However, in Tr-J, the absence of apoptotic morphology or long branched polymer (recognized by H10 clone anti-PAR antibody) suggest that the excessive nuclear PAR was not associated with a DNA repair response.

Our current work underscores the existence of a comparable increase of cytoplasmic PAR in a Charcot-Marie-Tooth model. Interestingly, the first scientific work that communicates an increase in human brain poly(ADP-ribosyl)ation with neurodegeneration (*Love, Barber & Wilcock, 1999*) is focused on the strong nuclear PARP and PAR signal in Alzheimer's disease patient brains. Such work does also show the existence of cytoplasmic PARP and PAR immunoreactivity, which is not further remarked. Other authors measure a significant PARP activity increase in hippocampus homogenates in contexts of excitotoxicity or amyloid beta peptide presence or, conversely, study the effect of PARP inhibition with 3-aminobenzamide (3-AB) (*Strosznajder, Jesko & Strosznajder, 2000*; *Strosznajder, Jesko & Zambrzycka, 2005*). They also arrive at the conclusion that PARP has a central role in neurodegeneration. We agree with that conclusion but in our opinion it has to be emphasized that neither PARP activity measurements nor PARP inhibition with 3-AB are evidencing exclusively nuclear PARP or PARP-1 activity. These results could be reflecting the activity and the role of a pool of PARPs (including PARP-1, PARP-2, TNKS-1 and TNKS-2), some of which are cytoplasmic.

In contrast to WT, Tr-J sciatic nerve axons harbor a strong PAR signal. PAR presence in peripheral axons resembles the work of *Brochier et al. (2015)* in the central nervous system, since they detect PAR in crushed optic nerve axons. Of note, we did not see any obvious difference in PAR signal intensity at the nerve cutting ends (we skipped them for quantification purposes just in case). *Brochier et al. (2015)* do also demonstrate that axonal growth inhibiting signals such as Nogo, MAG or astrocyte-produced chondroitin sulfate proteoglycans (CSPGs), induce neuronal PAR accumulation in primary cortical neurons. Likewise, the increase of MAG expression in the peripheral nervous system under neurodegenerative conditions (*Kinter et al., 2013*) might be related to the high levels of PAR in the Tr-J nerve fibers.

About 80% of the basal PAR pool is conserved in PARP-1$^{-/-}$ primary cortical neurons (*Brochier et al., 2015*). In contrast, the induced PAR increase depends on PARP-1 activity. PARP-1 is a nuclear enzyme; just a small PARP-1 fraction has rarely been localized in the cytoplasm of cancer cells (*Donizy et al., 2014*). Therefore, in Brochier et al. experiments, either neuronal PARP-1 can somehow reach the cytoplasm under certain conditions or PARP-1 is part of a signaling cascade that activates a cytoplasmic PARP that in turn synthesizes the cytoplasmic PAR. Interestingly, the PARP inhibitors which restore the axonal growth that is diminished by Nogo, MAG or CSPGs are not PARP-1 specific. Moreover, according to the same authors (*Brochier et al., 2015*), in a microfluidic-based culture platform, most growth rate restoration by the PARP inhibitor is achieved through PJ34 administration in the axonal compartment, sugesting that most PJ34 effects can be

reached through the inhibition of a cytoplasmic PARP. No increase in axonal regeneration nor improvement in motor function recovery were observed after optic nerve crush or dorsal hemisection of the spinal cord, in PARP-1 $^{-/-}$ mice or after systemic administration of the specific PARP-1 inhibitor velaparib (*Wang et al., 2016*).

Disturbances in PMP22 are associated with abnormal myelination in a range of inherited peripheral neuropathies both in mice and humans (*Robertson et al., 1997*). PMP22 is critical for actin-mediated cellular functions (*Lee et al., 2014*). Interestingly, actin microfilaments assembly and disassembly is essential during myelin sheath formation in the peripheral and central nervous system (*Park & Feltri, 2011*; *Feltri, Suter & Relvas, 2008*; *Nawaz et al., 2015*; *Zuchero et al., 2015*). A lamellipodia-like structure driving myelin wrapping has been described in peripheral and central myelination process (*Salzer, 2012*; *Feltri, Poitelon & Previtali, 2015*; *Nawaz et al., 2015*; *Zuchero et al., 2015*). Besides, PMP22-deficient nerves depict early abnormal junctions and permeability of myelin (*Guo et al., 2014*). Tr-J mice sciatic nerve has marked alterations in junctional proteins including delocalized E-cadherin (*Devaux & Scherer, 2005*). In turn, cumulative evidence indicates that *adherens junctions* proteins (E-cadherin, catenins, vinculin), some of which can act as NACos, may play a significant role in the myelination process (*Tricaud et al., 2005*; *Perrin-Tricaud, Rutishauser & Tricaud, 2007*; *Ye et al., 2009*; *Peng et al., 2010*; *Beppu et al., 2015*; *Basak et al., 2015*). For example, E-cadherin enhances neuroregulin-1 (NRG1) signaling and promotes Schwann cell myelination (*Basak et al., 2015*). It has to be emphasized that NRG1 is one of the major and best characterized extrinsic signals that control myelination (*Salzer, 2015*). Conversely, an aberrant localization of E-cadherin can be a potent inhibitor of Wnt/ β-catenin (*Su et al., 2015*) and in mammalian CNS, dysregulation of the Wnt pathway inhibits timely myelination (*Fancy et al., 2009*; *Dai et al., 2014*).

Considering our previous results regarding the actin cytoskeleton/PAR belt dialogue in VERO cells (*Lafon-Hughes et al., 2014*) and the highly increased actin observed in the Tr-J sciatic nerve (*Kun et al., 2012a*), the observed PAR increase was an expected result. Moreover, PAR-digested regions seemed to harbor lower F-actin signals, suggesting that PAR might be somehow participating in the holding or anchorage of at least part of the F-actin cytoskeleton network. In the nerves, like in VERO cells, the actin-PAR connection seems to be active.

We don't know if PARylation alterations are upstream actin and E-cadherin modifications. It is even unknown which proteins are PARylated in the sciatic nerve. Looking for putative cytoplasmic PARylation candidates –*adherens junctions*/actin anchorage proteins or microfilaments components-, we have inspected the macrodomain-recognized ADP-ribosylome of liver epithelium; catenin (CTNNA 3), vinculin, and β-actin are in the list of affinity-enriched proteins. As the macrodomain recognizes MAR and in some cases PAR (*Martello et al., 2016*), the method does not differentiate mono-ADP-rybosylated (MARylated) from PARylated proteins. Interestingly, this data fit with another proteomics work showing that in human HEK 293 embryonic kidney cells (*Gagné et al., 2012*), vinculin and catenin (β-catenin) were recovered in the pool of proteins bound to a catalytically inactive GFP-PARG or "PARG-DEAD domain" (indicating their probable PARylation or association to PARG) but not immunoprecipitated with clone H10 antibody

(indicating that they do not bind long branched chains). Actin was not recovered in non-stimulated cells. Instead, β-actin was enriched in PARylated complexes in MNNG-treated cells, indicating its participation in the responses induced by this alkylating agent (*Gagné et al., 2012*). In our knowledge, a single group claimed that actin is "the unique endogenous acceptor of PAR" in an *Octopus* brain cytoplasmic subcellular fraction (*De Maio et al., 2013*).

Monomeric G-actin has been long recognized as a MARylating target of bacterial toxins, shifting the equilibrium towards microfilament disassembly (*De Maio et al., 2013*). Coherently, the knockdown of PARP-14, which harbors MARylating activity (*Vyas et al., 2005*) and is a focal adhesion protein, results in cells that are unable to retract protrussions efficiently, generating highly elongated extensions (*Vyas et al., 2013*). The knockdown of the macrodomain-containing enzymatically-inactive PARP-9 results in abnormal membrane blebbing in the absence of typical apoptotic nuclear DNA hypercondensation (*Vyas et al., 2013*), suggesting that PARP-9 is also involved in actin cytoskeleton dynamics. As PARP-9 is enzymatically inactive and PARP-14 has just MARylating activity (*Vyas et al., 2005*), neither of them could be responsible for the synthesis of the PAR that we observed in mice sciatic nerves. Nevertheless, the existence of some sort of regulatory interaction among PARP-9 or PARP-14 and the PARP responsible for PAR synthesis cannot be discarded.

The analogy with epithelial cells (*Yeh et al., 2006*; *Lafon-Hughes et al., 2014*), points to TNKS involvement in cytoplasmic PARylation. Mice deficient in either one TNKS are viable but deficiency of both TNKS results in embryonic lethality, demonstrating that TNKS are essential but at least partially redundant (*Chiang et al., 2008*). Accordingly, TNKS-2 knockdown cells display no detectable phenotype whereas TNKS-1 knockdown cells display mitotic defects and diminished viability (*Vyas et al., 2013*), probably preventing further dissection of the molecular mechanisms involved.

Interestingly, a study on structural basis and sequence rules for substrate recognition by TNKS identifies a TNKS-binding motif (*Guettler et al., 2011*) which is present in vinculin and catenin but not in β-actin.

TNKS is necessary for canonical Wnt signaling (*Kartner et al., 2010*). TNKS inhibition induces axin stabilization and blocks Wnt signaling (*Bao et al., 2012*). Besides, axin has been identified as a regulatory and therapeutic target in newborn brain injury and remyelination (*Fancy et al., 2011*). Moreover, anti-TNKS weapons promote myelination (*Casaccia, 2012*). To be more precise, the TNKS inhibitor XAV939 accelerates oligodendrocyte progenitor differentiation in cell cultures, improves myelination and remyelination (following hypoxia or lysolecithin) in *ex vivo* mouse cerebellar slice cultures and diminishes the demyelinating effects of lysolecithin in mice spinal cord *in vivo* (*Fancy et al., 2011*). It has been demonstrated that the TNKS PARylation target molecule axin is involved. It is likely that some of the *adherens junctions*/actin anchorage proteins are PARylation targets too, acting in concert with axin to coordinate cell adhesion and migration with differentiation.

Together our findings highlight the presence of PAR in specific regions of peripheral nerves and its increase in Tr-J, opening a window to further explore the possible roles of cytoplasmic PAR associated to *adherens junctions*/actin cytoskeleton in the whole nervous system. The advances in this field are expected to contribute in the future to a more precise

design of therapies, not only for CMT patients, which represents 1 in every 2500 persons in the general population (*Li et al., 2013*), but also for other neurodegenerative disease patients.

## CONCLUSIONS

- PAR was detected in mice sciatic nerves, colocalizing with F-actin at non-compact myelin regions of peripheral nerve fibers which are rich in *adherens junctions*. The existence of PAR in *adherens junctions* regions in systems as divergent as Schwann cells and VERO epithelial cells suggests that PAR may be a previously overlooked inherent component of the E-cadherin rich *adherens junctions*.
- PAR was in excess in a Charcot-Marie-Tooth demyelinating disease model in which sciatic nerves present an excess of F-actin. Like in VERO cells, our result argue in favor of a structural (direct or indirect) connection of PAR with F-actin that deserves further investigation.
- Nuclear PAR was present in WT and moderately increased in Tr-J Schwann cells, putatively affecting chromatin structure and functions.

## ACKNOWLEDGEMENTS

We kindly acknowledge Archana Dhasarathy (University of North Dakota, USA) for the initial aliquots of ENZO anti-PAR antibody, Mariana Di Domenico (LEICA technician at Facultad de Medicina) for technical advice and Martín Breijo, Mariela Santos (URBE Facultad de Medicina), Carmen Pérez (Rodents Bioterio, IIBCE) for animal handling and maintenance and Sergei Nechaev (University of North Dakota, USA), for manuscript critical reading.

### Funding

This work was supported by the Comisión Sectorial de Investigación Científica (CSIC-I+D, C-213), Facultad de Ciencias, Universidad de la República, Uruguay; Programa de Desarrollo de las Ciencias Básicas (PEDECIBA, MEC, Universidad de la República), Uruguay; Sistema Nacional de Investigadores, Agencia Nacional de Investigación e Innovación (SNI-ANII), Uruguay; National Scientific and Technical Research Council (CONICET), University of Buenos Aires, Argentina; Agencia Nacional de Promoción Científica y Tecnológica (ANPCyT), Argentina. The funders had no role in study design, data collection and analysis, decision to publish, or preparation of the manuscript.

### Grant Disclosures

The following grant information was disclosed by the authors:
Comisión Sectorial de Investigación Científica: CSIC-I+D, C-213.
Facultad de Ciencias, Universidad de la República.

Programa de Desarrollo de las Ciencias Básicas (PEDECIBA, MEC, Universidad de la República).

Sistema Nacional de Investigadores, Agencia Nacional de Investigación e Innovación (SNI-ANII).

National Scientific and Technical Research Council (CONICET), University of Buenos Aires.

Agencia Nacional de Promoción Científica y Tecnológica (ANPCyT).

## Competing Interests

The authors declare there are no competing interests.

## Author Contributions

- Laura I. Lafon Hughes conceived and designed the experiments, performed the experiments, analyzed the data, contributed reagents/materials/analysis tools, wrote the paper, prepared figures and/or tables, reviewed drafts of the paper.
- Carlos J. Romeo Cardeillac conceived and designed the experiments, performed the experiments, analyzed the data, reviewed drafts of the paper.
- Karina B. Cal Castillo performed the experiments, analyzed the data, reviewed drafts of the paper.
- Salomé C. Vilchez Larrea and Silvia H. Fernández Villamil conceived and designed the experiments, analyzed the data, contributed reagents/materials/analysis tools, reviewed drafts of the paper.
- José R. Sotelo Sosa analyzed the data, reviewed drafts of the paper.
- Gustavo A. Folle Ungo analyzed the data, contributed reagents/materials/analysis tools, reviewed drafts of the paper.
- Alejandra E. Kun González conceived and designed the experiments, performed the experiments, analyzed the data, contributed reagents/materials/analysis tools, prepared figures and/or tables, reviewed drafts of the paper.

## Animal Ethics

The following information was supplied relating to ethical approvals (i.e., approving body and any reference numbers):

Institutional and national guidelines for the care and use of laboratory animals were followed. The experimental procedures were authorized by Comité de Etica en el Uso de Animales (CEUA)-IIBCE.

Protocol register numbers are: 011/11/2014 and 002/05/2016.

## Data Availability

Project: PAR in WT and Tr-J sciatic nerves. Color coding was maintained in most experiments. We registered PAR signal (488) in green, phalloidin signal (543) in red and DAPi signal (405) in blue.

Lafon hughes, Laura (2017): Cold Spring Harbor Lab Meetings_POSTER. figshare. https://doi.org/10.6084/m9.figshare.4611670.v1

Lafon hughes, Laura (2017): Image cuantification on ROIs. figshare. https://doi.org/10.6084/m9.figshare.4496483.v1

Lafon hughes, Laura (2017): Raw quantitative data. figshare. https://doi.org/10.6084/m9.figshare.4604335.v1

Lafon hughes, Laura (2017): Quantification summary. figshare. https://doi.org/10.6084/m9.figshare.4604347.v1

Lafon hughes, Laura (2017): Some digestion summaries per experiment in ppt. figshare. https://doi.org/10.6084/m9.figshare.4604350.v1 Retrieved: 17 53, Apr 20, 2017 (GMT)

Files from LEICA microscope: Lafon hughes, Laura (2017): IT4_.lif from LEICA microscope. figshare. https://doi.org/10.6084/m9.figshare.4604368.v1

Lafon hughes, Laura (2017): IT7_PAR en ratones_sin y con Ca_stacks nervio 010615 .lif. figshare. https://doi.org/10.6084/m9.figshare.4604422.v1

Lafon hughes, Laura (2017): 160615 nervio IT8.lif. figshare. https://doi.org/10.6084/m9.figshare.4604425.v1

Lafon hughes, Laura (2017): DIgestion 7 from EXP IT13. LEICA register. figshare. https://doi.org/10.6084/m9.figshare.4609288.v1

Files from OLYMPUS Fluoview (IT4, FV originals): Lafon hughes, Laura (2017): IT4_ some fluoview originals. figshare. https://doi.org/10.6084/m9.figshare.4609306.v1

Lafon hughes, Laura (2017): IT5_ TrJ87_ OLD but ok. figshare. https://doi.org/10.6084/m9.figshare.4609330.v1

Lafon hughes, Laura (2017): IT5_ TrJ87_ OLD but ok. figshare. https://doi.org/10.6084/m9.figshare.4609330.v1

Lafon hughes, Laura (2017): IT6_originales FV 240515. figshare. https://doi.org/10.6084/m9.figshare.4609783.v1

Processed .tif files and ROIsets (IT6_examples): https://figshare.com/s/888a0020a98669113e1f

Lafon hughes, Laura (2017): IT6 examples stack and ROIset. figshare. https://doi.org/10.6084/m9.figshare.4610803.v1

Lafon hughes, Laura (2017): AUXILIAR_ FIG 1 and 2. figshare. https://doi.org/10.6084/m9.figshare.4612861.v1

Lafon hughes, Laura (2017): AUX FIG3 TrJ. figshare. https://doi.org/10.6084/m9.figshare.4614634.v1

OCT6 sample images (FV originals): Lafon hughes, Laura (2017): OCT6 cryosections. figshare. https://doi.org/10.6084/m9.figshare.4797874.v1

Lafon hughes, Laura; calkarina@gmail.com (2017): OCT6 teasing. figshare. https://doi.org/10.6084/m9.figshare.4797850.v1.

## Supplemental Information

Supplemental information for this article can be found online at http://dx.doi.org/10.7717/peerj.3318#supplemental-information.

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
