# Peer review of "Poly(ADP-ribosylation) is present in murine sciatic nerve fibers and is altered in a Charcot-Marie-Tooth-1E neurodegenerative model"

_PeerJ, doi:10.7717/peerj.3318_

## Round 0.1 · original submission · Minor Revisions

The submitted ms has now been examined by two expert reviewers, and both raised a number of concerns, each one of which the authors should thoroughly deal with.

Reviewer #1 raised mainly questions regarding ms structure and style, and I advise the authors to correct all points raised.

In turn, reviewer #2 raised a number of specific technical issues, which should be clarified, and/or corrected to allow publication.

Please refer to their comments below and follow their requests to the letter. When finished, please explain in your resubmission letter the respective changes and/or replies to all comments.

Reviewer 1 ·

Basic reporting

The manuscript is in general well-writen. However, some changes are advice to improve readability, mainly by using shorter sentences. Also, the authors should refrain from introducing results in the methods section (see lines 176-182), as well as discussion issues in the results section. The conclusions sections is not clear. Please, revise line 537; it is not clear whether they are discussing hypothesis or results.

Please, check that "sodium butirate" in line 160 is correct.

Experimental design

The research is original and within the scope of the journal. This manuscript represents the first report on the presence of PAR in sciatic nerves. The invesigation is well performed and according to technical and ethical standards. The methods are very well described.

Validity of the findings

Data are robust. At least 7 paired animals (wild type vs. Tr-J mice) are studied in different experiments. It would be of interest if the authors discuss further on the variability (or not) found within the different experiments performed.

Conclusions section should be revised to make it clear.

Reviewer 2 ·

Basic reporting

Generally speaking, the use of English is clear and professional throughout the manuscript; although there are occasional spelling and grammatical errors.
The introduction successfully outlines the context of the study and the aims are clearly stated. In particular, the authors draw attention to the possible implication of abnormal PARylation in three common neurodegenerative disorders (Alzheimer’s, Parkinson’s and Huntington’s diseases). They note that research has generally focussed on the nuclear roles of ADP-ribosylation (DNA repair, chromatin structure, transcription etc), despite the fact that most ADP-ribosyl Transferase family are present in the cytoplasm.
The structure confirms to the PeerJ standards.
The figures are generally of high quality, and are well described by the captions. It may be useful to include arrows highlighting notable regions, where appropriate.
The Raw data has been supplied in accordance with PeerJ policy.

Experimental design

The research is within the scope of the journal.
The research question is well defined, relevant and meaningful. The study draws attention to a previously overlooked field of PAR biology.
The methods are described in detail and contain sufficient information for replication.

Validity of the findings

Fig 1.
- The colocalization of the anti-PAR and phalloidin-543 signals is not evident from the image presented in Fig 1A. It would be useful to present split-channel images, so that the distribution of each signal can be clearly seen.
- The specific anti-PAR antibody should be detailed in the figure caption.

Fig 2.
- The authors report a 3-4 fold elevation in cytoplasmic anti-PAR and phalloidin-543 signals across the nerve diameter in Tr-J vs WT nerve sections. This observation appears to be sound. However, given that the authors note in the introduction that Tr-J mice have increased numbers of Schwann cells in their sciatic nerve fibres, it would be useful to have a control to demonstrate that the gross anatomical differences between the WT and Tr-J nerves has not skewed their signal quantification. For example, the authors could demonstrate that the relative signal of an unrelated cytoplasmic marker, such as GAPDH, does not increase in Tr-J vs WT nerves.
- The specific anti-PAR antibody should be detailed in the figure caption.

Fig 3.
The specific anti-PAR antibody should be detailed in the figure caption.

Fig S1.
- The authors present controls for the specificity of the signal detected by ENZO mice anti-PAR
antibody (BML-SA216). The use of PARG incubation to demonstrate signal specificity is commendable. However, similar controls are not presented for the BD anti-PAR antibody which is also used throughout the study, presumably including the main Figures 1,2 and 3. Given that multiple anti-PAR antibodies have been used in this study, these should be detailed in each figure caption.

Additional comments

- The authors have chosen to use an ImageJ colocalization plugin which highlights pixels in which anti-PAR and phalloidin-543 signals are greater than user-defined thresholds for each. A specific citation for this plugin would be useful, as a number of such plugins exist.
- Given that the study is presenting a localization of PAR in the adherens junctions, the data would be improved by the inclusion of images revealing the colocalization or not of anti-PAR and anti-E-cadherin directly, if possible.

---

## Round 0.2 · Minor Revisions

Please check the compound as requested by the reviewer.

Reviewer 1 ·

Basic reporting

The authors have adequately addressed most of the concerns raised by reviewers.

Experimental design

The authors have adequately addressed most of the concerns raised by reviewers.

Validity of the findings

The authors have adequately addressed most of the concerns raised by reviewers.

Additional comments

Please, check again: NaBH4 corresponds to sodium borohydride, not to sodium butiyrate. These are completely different things.

---

## Round 0.3 · accepted · Accept

Revised manuscript is now accepted for publication in PeerJ.